# The Relationship between Perceptions and Objective Measures of Greenness

**DOI:** 10.3390/ijerph192316317

**Published:** 2022-12-06

**Authors:** Joy L. Hart, Ray A. Yeager, Daniel W. Riggs, Daniel Fleischer, Ugochukwu Owolabi, Kandi L. Walker, Aruni Bhatnagar, Rachel J. Keith

**Affiliations:** 1Department of Communication, College of Arts and Sciences, University of Louisville, Louisville, KY 40292, USA; 2Christina Lee Brown Envirome Institute, University of Louisville, Louisville, KY 40202, USA; 3Division of Environmental Medicine, School of Medicine, University of Louisville, Louisville, KY 40202, USA; 4Hyphae Design Laboratory, Oakland, CA 94607, USA

**Keywords:** perceptions of greenness, normalized difference vegetation index, tree canopy, leaf area index

## Abstract

Exposure to greenness has been studied through objective measures of remote visualization of greenspace; however, the link to how individuals interpret spaces as green is missing. We examined the associations between three objective greenspace measures with perceptions of greenness. We used a subsample (n = 175; 2018–2019) from an environmental cardiovascular risk cohort to investigate perceptions of residential greenness. Participants completed a 17-item survey electronically. Objective measurements of greenness within 300 m buffer around participants home included normalized difference vegetation index (NDVI), tree canopy and leaf area index. Principal component analysis reduced the perceived greenspaces to three dimensions reflecting natural vegetation, tree cover and built greenspace such as parks. Our results suggest significant positive associations between NDVI, tree canopy and leaf area and perceived greenness reflecting playgrounds; also, associations between tree canopy and perceived greenness reflecting tree cover. These findings indicate that the most used objective greenness measure, NDVI, as well as tree canopy and leaf area may most align with perceptions of parks, whereas tree canopy alone captures individuals’ perceptions of tree cover. This highlights the need for research to understand the complexity of green metrics and careful interpretation of data based on the use of subjective or objective measures of greenness.

## 1. Introduction

Exposure to greenness has been associated with several positive health outcomes ranging from psychological effects, such as reduced stress [1] and feelings of anxiety [2], to physiological effects, such as reduced blood pressure [3] and inflammation [4], as well as reduced morbidity and mortality [5,6]. Such salubrious health outcomes have been reported in epidemiological studies as well as so called “forest bathing” studies, in which individuals are immersed in greenspace or forests. Despite several reviews and meta-analyses [7,8,9,10,11] that support the beneficial impact of greenness, there is sparse causative data to date and indices for greenness vary from study to study. However, whether there are casual relationships between exposure to greenness and various mental states [12], the underlying mechanisms remain unclear. For instance, it is unclear how greenness affects mental states and whether these effects are entirely psychological or due to some other effects of greenness, such as sunlight [13], reduction in noise [14], light [15], or air pollution [16], or due to plant-derived volatile chemicals that can affect psychological states [17]. Hence, to fully understand the health effects of greenness and to guide the development of greenness interventions to improve health, a better understanding of the mediators of the health effects of greenness is required.

To understand how greenness affects health, it is important first to understand what greenness means. Objective measures of greenness are standard over different spaces and geographic locales, allowing their use with large datasets, such as those in epidemiological studies, to assess health and greenness. Many large epidemiological studies use a common objective measure of a remote sensing-based index of greenness called normalized difference vegetation index (NDVI). NDVI can be aggregated into different spatial resolutions and longer-term averages of vegetation yield a good estimate of overall neighborhood greenness when compared to expert-rated perception of greenspace [18]. Previous findings indicate a relationship between health and green space anywhere from 50 m to 3 km radius from an individual’s home [19], but there is debate about the measure when it comes to spatial resolution and variability in relation to urban areas [20]. Another method to measure greenness is Light Detection and Ranging (LiDAR). LiDAR images can be processed to provide other metrics of high-resolution, such as leaf area, canopy volume and biomass, providing the most detailed account of non-grass and shrub greenness as well as greenness due to trees. For speciation from remote sensing techniques, validation of species in a particular area through an audit system can be employed. These measures may be important because diversity in the types of greenness is thought to provide additional health benefits [21,22,23].

Leaf area index (LAI) is a dimensionless measure of the amount of vegetative material in a canopy as the ratio of one-sided leaf area per unit ground area for flat leaves or half the total intercepting area for non-flat leaves [24]. LAI is difficult to measure accurately and difficult to quantify due to the differences seen in spatial and temporal measures [25]. Given the technical difficulty associated with such measures, LAI is not readily available for many health studies. However, because of its high resolution and temporal sensitivity, LAI could be a valuable in health studies as it may better account for the total volume of the vegetative mass which would be important for air pollution, noise, and light buffering, all of which can affect health outcomes.

Nevertheless, LAI as well as other objective measurements of greenness do not consider the individual’s perspective in evaluating surrounding greenness. Surveying individuals can provide more subjective insights on greenness and can assess how “green” an individual believes their environment is based on their perceptions of specific natural elements. Such survey measures consider individual perspectives, which may vary for the same geographical space (i.e., based on lived experience ultimately allowing different perceptions of greenness for the same space). Given increased interest in how greenness relates to health, research examining how perceptions of greenness are associated with objective measures of greenness is emerging. Some outcome studies suggest a benefit to health when looking at both objective and subjective measures of greenness [26], whereas others suggest perceptions of greenness are a better measure for research on health effects [27]. Understanding the benefits of each type of objective and subjective measure of greenness is important. This emerging body of literature suggests that the utilization of disparate remotely-sensed measures [28] or the differences in spatial buffers compound these relations [29,30]. One study found that tree canopy within a 10 min walk is associated with perceived greenness [31]. Whereas research conducted in Mexico [32] and Australia [28] as well as a systemic review [29] suggest that common objective measures are not reflective of perceptions of greenness. Given that there may be different outcomes based on country of origin [30], perceptions from different geographical areas and cultures may influence findings. Importantly, perceptions of greenness may have more significant associations with health than objective measures that are more commonly used in large epidemiologic health studies [33].

To understand the health effects of greenness, careful consideration should be given to how greenness is defined. A recent systemic review by Bowler et al. (2010) suggested many health studies use areas like parks and university campuses with participants immersing themselves in these spaces to evaluate the health effects of greenness and were selected based on the authors’ descriptions that they were ‘relatively green’ [10]. Of the studies reviewed, most involved short durations of one-hour or less [10], limiting the generalizability of the information collected. Of note, a recent systemic review by Rigolon et al. (2021) suggested spaces like parks had stronger protective effects than general green cover for lower socioeconomic status populations [30]. Tzoulas et al. (2007) published a literature review specifically examining green infrastructure in urban areas and its contributions to human health [9]. This more comprehensive view of greenness defined urban green infrastructure as encompassing the array (i.e., natural, semi-natural, and artificial) of networked ecological systems existing in and across areas regardless of spatial scale [9,34]. Multiple beneficial health endpoints included longevity, increased well-being and more positive mental health through various proposed mechanisms including decreased air pollution, increased physical activity, and socioeconomic status [9]. Other work has suggested greenness in forested vs. urban areas may have differential health effects [35]. Though many tout the benefits of greenness, some studies have shown a negative relationship between greenness and health. One such study focused on the aesthetics of greenness suggesting that overgrown or unmanaged areas could increase anxiety associated with a fear of crime [36] leading some to prefer more wild spaces or built environments instead of urban greenspaces [37]. Despite this possible perception, there is data suggesting that urban greenspace can decrease crime [38] and that individuals with lower socioeconomic status who access parks that may not be as well-kept do have beneficial health outcomes [30]. Some also associate greenspaces with the potential for vector borne infectious disease [39]. These negative perceptions could also influence how greenness effects mental health.

Because individual health is the totality of both physical and mental health, including perceptions of health, how an individual perceives their surroundings (e.g., level of greenness) may be important in terms of health outcomes. Extensive research suggests that an individual’s perceptions of their health behaviors, independent of the behavior itself, is sufficient to influence physiological responses [40] and predict mortality [41]. For example, in one study using stressed participants, those who believed that high levels of stress would affect their health had an increased risk of premature death [42]. Additionally, perceptions of health benefits, such as weight loss, decreased blood pressure and reduced body fat, have been linked to the effects of such behaviors on health [43]. Taken together such findings suggest that perception of wellness can be as important to health as the behavior itself. Hence, when evaluating the effect of greenness on health, it may be prudent to understand an individual’s perception of greenness. There are different ways to measure perceptions of greenness just as there are different ways to measure objective greenness. Knobel et al. (2021) measured perceived access to greenspace in relation to cardiovascular risk factors and found that socioeconomic, race and ethnic background changed benefits seen from the greenspace [44]. Others have looked at how aesthetic qualities, such as overgrown areas, may be perceived in relation to the effects on health [36,45]. Simple scales also have been employed to characterize dimensions of urban environments most people associate with greenness [28]. How greening affects perceptions of the overall environment, not just the greenspace in the environment, is also captured by self-report [46]. A systemic review by Knobel et al. (2019) suggested that a comprehensive definition of greening is lacking and who uses these measures of greenness (i.e., experts or community) also varies. It also noted great variability among the different measures of greenness making them hard to compare [47] and leading to larger problems with generalizability of results across studies.

Given increased interest in how greenness can influence health, careful consideration should be devoted to choosing the most representative measure of neighborhood greenness when designing green health studies [48]. Previous work suggests that NDVI, a commonly used objective measure of greenness, may not accurately reflect an individual’s perception of greenness around their residence [28]. Accordingly, a person’s perception of their environment may not correlate with the objective measures used by researchers to determine neighborhood greenness levels. Additionally, given discordance between green measures and perceived measures of health as well as the fact that most current greenness and health data are derived from large epidemiological studies that do not include perceptions of greenness, it is important to determine how objective measures of greenness are correlated with a perceptual measure of greenness. Our hypothesis is that perceived greenness will increase with higher levels of natural vegetation and trees, and these dimensions will associate most with tree canopy or leaf area index. Thus, we designed this study to examine the correlation between objective measures of greenness with perceptions of greenness to provide guidance on choosing the best measure for larger health and epidemiological studies. 

## 2. Materials and Methods

### 2.1. Participants

Participants from an ongoing cardiovascular risk cohort were recruited to participate in an online survey. All study-related procedures and measures were approved by the University of Louisville’s Institutional Review Board (IRB #15.126), and informed consent was obtained prior to data collection. Of 733 participants invited in November 2019 to complete the approximately 20 min online survey of their perceptions of the greenness around their residence in South Louisville (Kentucky) (IRB# 18.1227), 175 responded to the perceptions surveys (Figure 1). All participants lived within an approximately 4-mile study area. Tree canopy, leaf area and NDVI measures were completed for the full study area and used as objective measures of greenness within a 300 m round radius based on participant residential address. 

For this study of perceptions of greenness, 201 participants were recruited, and 175 satisfied the inclusion criteria (n = 26 excluded due to >50% missing questionnaire data). Figure 1.

### 2.2. Subjective Greenness Measures

Perceptions of greenness were measured using a 17-item scale previously used by Leslie et al. [28] and supplemented with additional questions about satisfaction with the levels of greenness (Appendix B). Responses were collected and managed using Research Electronic Data Capture (REDCap) electronic data capture tools hosted at the University of Louisville [49,50]. The questions surveyed participants about their residential surroundings and were scaled from 1 to 4 (1 = strongly disagree through 4 = strongly agree), with 1 item reversed scored. The higher the individual rated each item or the higher the summed score of all items, the higher the level of perceived greenness. All survey responses were collected over a 2-week span. Responses to 17 self-reported perception of greenness items and inter-item Spearman’s correlation are reported.

### 2.3. Objective Greenness Measures

We calculated NDVI with cloud-free multiband images retrieved from Planet Lab’s Planetscope at 4 m resolution. The images were acquired from Planetscope 0 (dove-classic/dove/PS2) and Planetscope 1 (dove-R/PS2.SD) generations of satellites, with Planetscope 0 images adjusted to the Planetscope 1 scale using the normalization coefficients published by Huang and Roy (2021) [51]. We calculated the mean NDVI of summer images from throughout the cohort enrollment period in 2018 and 2019 in order to correct for any date-specific artifacts in a single-image dataset, such as residual clouds, season phenology, or recent weather events such as excess rainfall or drought.

We calculated both canopy and LAI from aerial-based LiDAR imaging gathered from 15 August 2019, to 17 August 2019, in 3 flights by Quantum Spatial Inc. (Lexington, KY, USA) using a Leica ALS70 LiDAR sensor flown at an altitude between 1167 and 1270 m, at a ground-speed of 120 kts. The sensor was set at a scan rate of 69.5 Hz, and a laser pulse rate of 221.1 kHz. With a 15° field of view and 334 m wide swaths spaced 150.3 m apart, the resulting point cloud density averaged 19.77 points per m^2^. 4-band imagery was acquired by Quantum Spatial on September 9th, 2019, in a single flight at 12,500 ft of altitude at a ground speed of 150 kts, using an UltraCam Eagle RGB-NIR sensor with 100 mm focal length. Imagery swaths were 16,400 ft wide with overlap of 55%, yielding a ground sampling resolution of 0.6 ft. Data was rasterized at 1 m^2^. We estimated LAI from these aerial LIDAR data using variations of the Beer-Lambert law, according to the equations published by Klingberg et al. (2017) [52]. Canopy area was determined by using supervised machine learning classification with LiDAR and multispectral data to delineate areas of canopy cover.

We compiled the mean value of NDVI, the total leaf area, and the percent canopy coverage within radial buffer areas of 300 m by applying the Focal Statistics tool in ESRI ArcGIS Pro, with rasterized greenness datasets as the input and a raster dataset of 1 m resolution cells representing each greenness dataset mean value of a buffer area within 300 m as the output. We then extracted the overlaying raster cell value at geocoded participant residential point locations, which represented the mean value for each greenness metric within a 300 m spatial buffer area. For participants residing in large multiunit housing addresses, we adjusted geocoded point locations to reflect the unit location within the housing complex prior to greenness data extraction. We manually examined and verified extracted values for accuracy before finalization and analysis.

### 2.4. Statistical Analysis

Baseline participant characteristics were stratified by tertiles of residential NDVI and tree canopy values. NDVI was classified as low (≤0.36), medium (0.37 to 0.40) and high (>0.40); tree canopy was classified as low (≤25.00), medium (25.1 to 29.5) and high (>29.5). Frequencies and percentages were reported for categorical variables; means and standard deviations were reported for continuous variables. Differences in participant characteristics by tertiles of greenness metrics were tested using Chi-square test for categorical variables and one-way ANOVA for continuous variables (Table 1). *p*-values ≤ 0.05 were considered statistically significant.

Spearman correlations were estimated between the 17 questionnaire items with NDVI, leaf area, and tree canopy (Table 2). Principal component analysis (PCA) with varimax rotation was used for dimensionality reduction of the 17 questionnaire items. The first 3 components with eigenvalues greater than one were used for further analysis. Linear regression models were used to estimate the relationships between the 3 principal components with: NDVI within a 300 m radius, leaf area within a 300 m radius, and tree canopy within a 300 m radius. Models were adjusted a priori for annual household income category (<$20,000, $20,000–$64,999, $65,000–$124,999, >$125,000), race (White, Black, other), education (high school graduate or below, college graduate or some college, graduate school), and sex (male, female). Results are reported as adjusted Beta per IQR of greenness metric. To account for multiple testing between the 3 principal components and metric of greenness, we applied the Benjamini-Hochberg (BH) procedure to control the false-discovery rate. The adjusted alpha was determined as 0.028 after applying the BH procedure. All analyses were performed using SAS version 9.4 software (SAS Institute, Inc., Cary, NC, USA).

## 3. Results

### 3.1. Study Characteristics

As reported in Table 1, most participants were White and female. About half had incomes of $20,000–$64,000, and 28% had household incomes of $65,000–$124,999. Significantly different covariates between residents that lived in high, medium, and low 300-m-radius NDVI levels include higher income in the greenest areas and higher weight and waist circumference in the least green areas. When the cohort was stratified based on high, medium, and low 300-m-radius tree canopy levels, we found higher levels of income and education in the greenest areas and a higher percentage of participants with diabetes in the areas of lowest greenness (Appendix A).

### 3.2. Objective Measures of Greenness

As shown in Figure 2, NDVI at a 300 m buffer was significantly associated with tree canopy at a 300 m buffer (R^2^ = 0.764, *p* < 0.001) and leaf area index at a 300 m buffer (R^2^ = 0.800, *p* < 0.001). Tree canopy at a 300 m buffer was significantly associated with leaf area index at a 300 m buffer (R^2^ = 0.927, *p* < 0.001).

### 3.3. Perceptions of Greenness

Internal consistencies for the greenness items are reported as Cronbach’s alpha for all 17 items: 0.90 (95% CI: 0.87, 0.92). Cronbach’s alpha also was reported when poorly loading items were deleted from each dimension, respectively—Items with loadings on dimension 1, 0.86 (95% CI: 0.84, 0.89); Items with loadings on dimension 2, 0.79 (95% CI: 0.75, 0.84); Items with loadings on dimension 3, 0.86 (95% CI: 0.83, 0.89).

Most of the 17 perceived greenness items had mild to moderate inter-item correlations (Table 2). As described in Table 2, there are multiple significant bivariate correlations between perceived greenness and objective measures of greenness. Significant weak positive correlations with NDVI and leaf area include: Item 3: Local park or nature reserve close to where I live (r = 0.201; r = 0.177); Item 5: School grounds with grassed areas nearby (r = 0.179; r = 0.150); Item 6: Views of nature from my home (r = 0.258, r = 0.230); Item 7: Tree cover or canopy along walking routes (r = 0.151; r = 0.242); Item 8: Many large trees in my local area (r = 0.164; r = 0.258); Item 9: Many roadside plantings of trees and shrubs (r = 0.154; r = 0.246); Item 10: Many street trees in my local area (r = 0.166; r = 0.251); Item 11: Walking or bicycle paths or trails nearby (r = 0.235; r = 0.300); Item 12: Pleasant natural features (reserves, beach, lake); and Item 16: Lots of greenery around my local area (trees, bushes, gardens) (r = 0.167; r = 0.247). Additional significant weak correlations were seen for leaf area with Item 2: Children’s playground nearby (r = 0.220); Item 14: Lots of green median strips (r = 0.151) and Item 15: Lots of vegetation in nearby gardens (r = 0.157). Compared with NDVI and leaf area measures, tree canopy retained significant weak associations for all items except Item 3, Item 8, Item 12, Item 14 and Item 15.

### 3.4. Principal Component Analysis

Eigenvalues, percent of variance explained, and component loadings for the first three principal components are reported in Table 3. The first three dimensions explained 66% of the total variation. Dimension 1 is positively correlated with questions related to gardens, natural features, native vegetation, lots of vegetation and gardens and green median strips. Dimension 2 is positively correlated with questions related to tree cover or canopy, many large trees, and many street trees and negatively correlated with little or no lawn in yards. Dimension 3 is positively correlated with questions related to children’s playgrounds, local parks, and sports fields.

### 3.5. Associations of Perceived Greenness and Measures of Objective Greenness

In adjusted models, NDVI within a 300 m buffer around participant residence has a positive association with dimension 3 (β = 0.265 per IQR of NDVI; 95% CI = 0.092, 0.438; *adjusted p* = 0.026) (Table 3). There were no other significant associations between dimensions and NDVI in adjusted models (Table 4). 

In adjusted models, tree canopy (TC) within a 300 m buffer around participant residence has a positive association with dimension 2 (β = 0.300 per IQR of TC; 95% CI = 0.078, 0.523; adjusted *p* = 0.026) and dimension 3 (β = 0.296 per IQR of TC; 95% CI = 0.068, 0.524; adjusted *p* = 0.026) (Table 4).

In adjusted models, LAI within a 300 m buffer around participant residence has a positive association with dimension 3 (β = 0.280 per IQR of TC; 95% CI = 0.075, 0.485; *adjusted p* = 0.026) (Table 4).

## 4. Discussion

In this study, we compared self-reported perceived greenness with 3 objective measures of greenness—NDVI, tree canopy and leaf area within a 300-m-radius of the individuals’ residence to understand what objective greenness metrics were most associated with neighborhood attributes our cohort perceived as residential greenness. Tree canopy and leaf area index were highly correlated with NDVI. Despite these correlations there were differences between each objective measure and associations with the individual items assessed in the perceptions surveys as well as the 3 dimensions we found using the perceived measure. For individual items in the perception of greenness survey, leaf area index showed the highest association of all 3 measures and this association was with item 11—walking paths or trails nearby. The second highest correlation was seen with NDVI and item 6—views of greenness from my home. The third highest correlation was seen with LAI and item 11. Interestingly, the item that showed the highest correlation across all 3 measures was item 11 that addressed walking paths or trails nearby, followed by item 6 that addressed visible greenness from the home. Weakest correlations were seen for item 1—formal parks nearby, followed by item 4—sports fields nearby and this was consistent across all 3 measures. Item 17—little or no lawn nearby showed consistent negative associations and was the only inverse relation identified. Overall TC and LAI had the same number of significant relationships of individual items, and interestingly had more than NDVI. Despite the numerous associations found, all correlations were mild to moderate; therefore, the outcomes may not have generalizable practical importance. All 3 measures of greenness were associated with dimension 3, which described features of the built environment such as children’s playgrounds, local parks, and sports fields. Tree canopy was the only objective measure associated with dimension 2, representing trees. Interestingly leaf area was not associated with the dimensions that were most reflective of trees and leafy vegetation. There were no significant relationships found with dimension 1, though we did find a trend toward a possible relationship with NDVI and tree canopy that should be explored with future research. Most importantly, though, in our sample there were no significant relationships between objective measure that captured the perceptions of general vegetation, or dimension 1, which our cohort identified as what they believed most represented “greenness”. These findings illustrate that our study’s objective measures of greenness, even when measuring specific greenness attributes such as tree canopy or leaf area, do not accurately represent perceived greenness, as characterized by natural spaces, natural vegetation, gardens, and green medians. Our results underscore the findings of Leslie et al. regarding remaining conceptual challenges in classifying and measuring greenness [28].

NDVI is an objective measure of greenness commonly used in epidemiology to examine associations between greenness and health. In our study, NDVI was only associated with dimension 3, which accounted for the smallest section of perceived greenness and most reflected spaces that were open and highly maintained such as children’s playgrounds, parks, and sports fields. Many links between greenness and health have been found in such large-scale studies with few studies exploring other objective or subjective measures of greenness. NDVI may be the tool of choice as it is relatively easy to acquire and a useful metric of vegetation coverage often associated with health outcomes; however, our work and that of others [28] suggests it does not accurately reflect how community members residing in the area perceive greenness, which in our case is more natural spaces. One factor that may contribute to a different perception of greenness from the objective measures is the use of different spatial resolution buffers. For instance, the associations of NDVI with health measures has been reported across different spatial resolution buffers, meaning that the associations with health outcomes could be examined with buffers ranging from 50–500 m distances. For example, a small buffer area around a person’s home may be most influential on perception when compared with a larger surrounding area.

Recent work suggests that using a larger buffer area tends to be associated with greater health outcomes [53]. However, residents may not think of features that are 500 m away when they are thinking of the greenness in their lived space. Additional studies suggest that the NDVI dataset itself that is used in a study can influence the associations with health [54]. Furthermore, NDVI data that do not match the season or year of outcomes data are often employed, further confounding interpretation of results. These issues suggest that researchers may not be using the best tool to identify the potential effects of green aesthetics that contribute to restorative effects [55] or understand the speciation that could expose individuals to physiologically active plant biologics [56]. Across studies, discordance in findings related to greenness and health continues [57], and this discordance may arise, in part, from how greenness is measured and differences in perceptions of that greenness.

To rigorously investigate the role that greenness plays in human health, many investigators have begun to look for more discrete or descriptive ways to measure greenness. Emerging research suggests, as with NDVI, there is evidence of associations between tree canopy and mental health [58] as well as physical health [59,60,61]. In our study, tree canopy was the only objective greenness measure associated with the perception of greenness most aligned to trees (dimension 2), yet it was also associated with dimension 3 and areas like playgrounds, parks or sports fields. Similar to our findings, an Australian study by Mazumdar and colleagues reported in 2020 that tree canopy within a 10 min walking buffer may be best associated with perceptions of greenspace [31]. However, this measure has rarely been assessed in epidemiological studies. Additionally, newer remote measures have been developed to assess what greenspaces may be visible from an address based on photography or field surveys and these tend to have a high correlation with tree canopy [47,62]. The hypothesis that the green you see may be more likely to influence your health has discordant findings with some studies suggesting that these “seen” green measures do not correlate well with other objective measures of greenness [32]. This may be in part because at least some of the health benefits from greenness may be associated more with trees than grasses or other greenspace [60], yet the most commonly used measure, NDVI, does not distinguish between types of greenness.

Although LAI is most often used to assess trees, forests and vegetative health, our data suggest, like NDVI, the perceptions of greenness most associated are dimension 3. To date, there has been little research on health outcomes and LAI. Both TC and LAI use spatial buffers, are subject to model bias and have the potential for misclassifications similar to NDVI, but our data suggest that tree canopy is associated with more dimensions of perceived greenness than LAI or NDVI. Independent of the objective measures of greenness, perceptions of greenness could drive the associations between health and greenness in several ways. For example, the aesthetics of the vegetative areas may provide a benefit through reduced stress, by a restorative effect or by encouragement of exercise [55]. Therefore, no matter how greenness is objectively measured, it may not reflect true exposure, if these exposures do not reflect an individual’s perception of greenness in the area. This mismatch between perceived and objective greenness could significantly alter the magnitude of potential health benefits. Additional thoughtful consideration of objective and perceived green metrics is a topic of critical importance in minority communities as appreciation of nature is culturally specific [63].

Previous work has shown that perceptions can be closely tied to health outcomes [40,41,43,64]. Perceptions of greenness can be fluid based on differences in the makeup of the natural environment. For example, some previous work that suggests the environment in which a study is conducted may lead to more discordance than the measures used to assess perceived and objective greenness [65]. The different relationships between the subjective and objective greenness scales that arose when using the same scales in different environments may be due to the variations in the types of greenness across different environments, suggesting the attributes of the greenness are important in perceptions but may not be captured in all objective measures. Our findings suggest that most of our participants thought of natural spaces, natural vegetation, gardens, and green medians as greenness but none of our objective measures were related to this dimension. Instead, our objective measures were most reflective of large open, often communal, greenspaces like playgrounds, parks and sporting fields. Given that this study, like many greening studies, was conducted in an urban population, access to natural vegetation and gardens within a 300-m-radius buffer may be limited. Additionally, if the objective measure of greenness, such as NDVI, does not match the study population’s perceptions about the space, the health effects assigned to greenness may miss those most influenced by aesthetics. Alternatively, if trees are influencing health in physical ways, such as reducing air pollution or giving off biologic chemicals, then perceived green measures alone could miss important health effects. Therefore, incorporating multiple objective and subjective measurements of greenness could provide a more comprehensive assessment of exposure to fully evaluate the impact of greenness and to delineate the mechanism underlying the salutary effects of greenness. 

This study has several strengths including multiple measures of objective greenness and a participant cohort in a small geographical area with all essentially exposed to similar spaces. We chose to use multiple measures of objective greenness because traditionally each measure has been used to describe different aspects of greenness. Specifically, NDVI is used for general vegetation density, TC is used specifically to calculate trees in the area and LAI may reflect the specific nature of the tree and bush canopies including volumes and types of trees. These measures were all obtained using high resolution data captured temporally to the collection of perceptions. The small geographical footprint allowed us to capture multiple perceptions for the same neighborhood space. Despite these strengths, our study has some limitations. Our questionnaires were completed during early fall and tree canopy might not have been at its peak. We have a limited sample distributed across a relatively small spatial scale, so despite strengths of study design, replication is needed for generalizability. Additionally, there could be residual confounding. Continued work employing both objective and subjective measures of greenness with urban populations is important to better inform public health.

Greenness has become popular in public health interventions and policy [12], yet we still do not have consensus on measurement, which limits how to intervene most appropriately in regard to health outcomes. Indeed, cities across the US and globe are rolling out initiatives to actively increase tree canopy—in part to increase health and decrease environmental inequities [66]. Therefore, it is important to understand both the strengths and weaknesses of currently available objective greenness measures as well as to consider perceptions of greenness when designing health outcome studies. To fully understand the outcomes of these interventions, we must continue to explore the relationship between objective and subjective measures of greenness. We must also start to evaluate different objective greenness metrics in health studies based on desired outcomes.

## 5. Conclusions

In summary, the data suggest that the particular objective green measure employed in studies may represent different aspect of greening in relation to perceptions. Our study cohort identified 3 components that they believed described greenness including natural vegetation, trees, and green features of built environments such as playgrounds, parks, and sports fields. Our objective measures only captured 1–2 of these components. There is also the potential to develop novel new green measures that could more accurately reflect the complexity of natural green spaces as a whole, or studies should consider incorporating multiple objective and subjective measures of greenness to most accurately reflect the space and its residents as none of the objective measures studied was able to capture that perceived green space. Moving forward, for health-related studies, more attention could be focused on which green metrics would be best, whether there can be harmonization of measures, and how data are interpreted based on different objective or subjective (perceptions) green measures.

## Figures and Tables

**Figure 1 ijerph-19-16317-f001:**
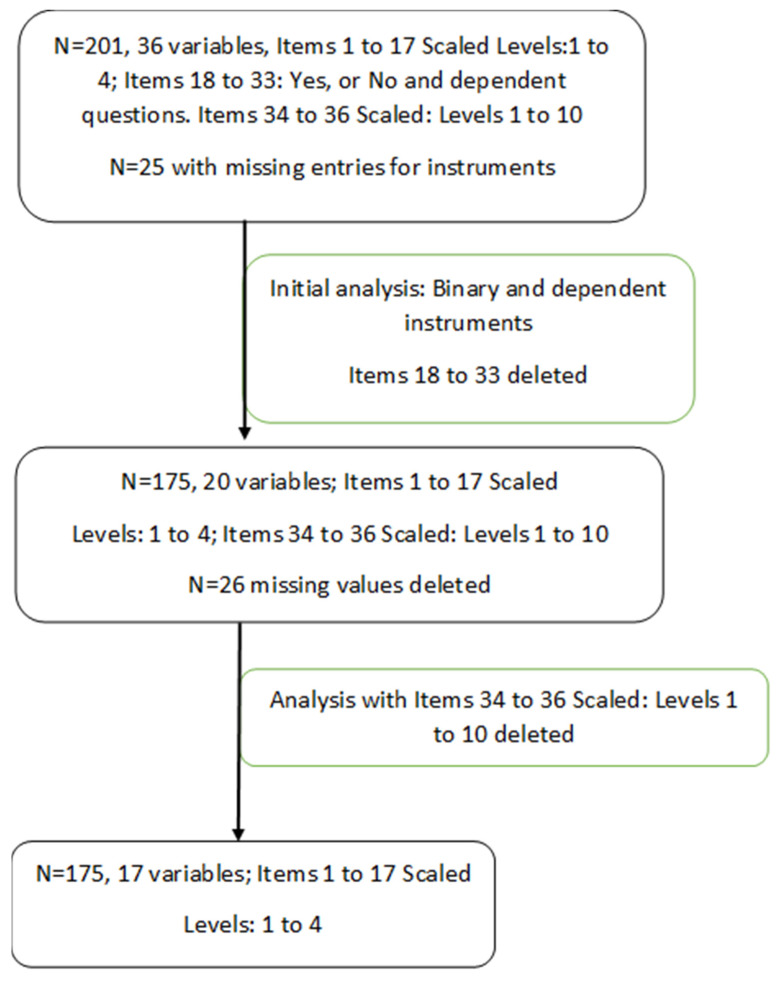
Study flow chart.

**Figure 2 ijerph-19-16317-f002:**
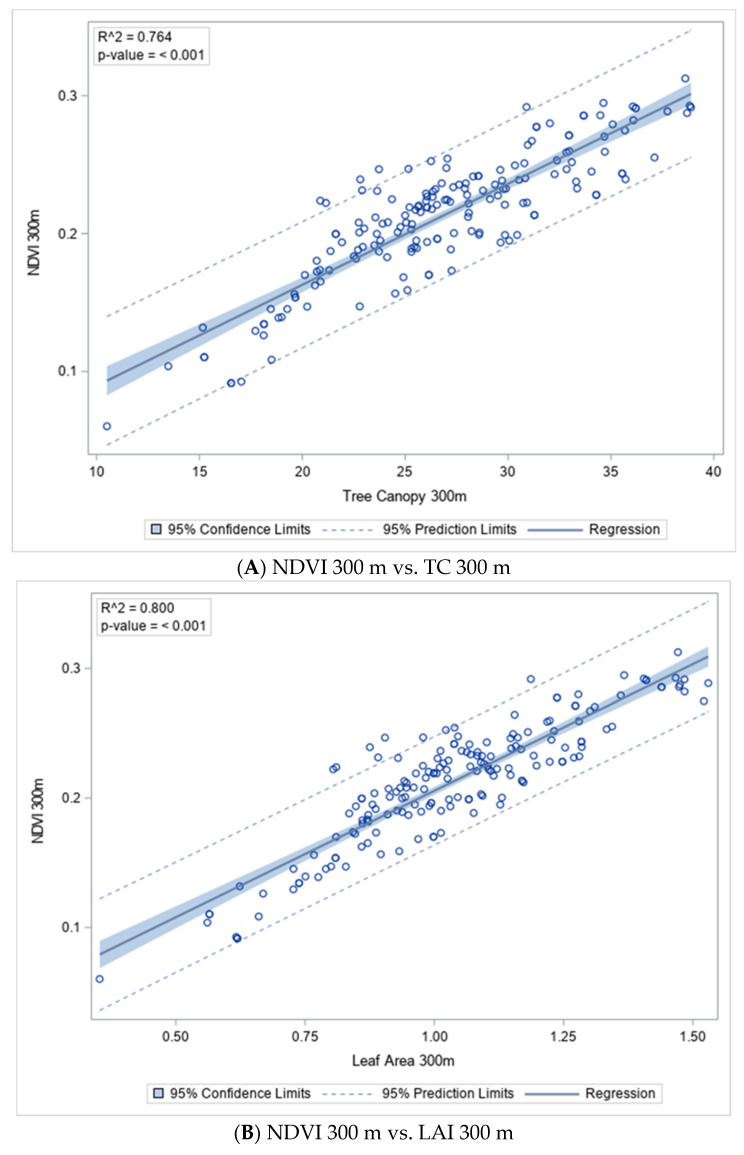
Unadjusted Linear regression models between (**A**) NDVI and Tree Canopy, (**B**) NDVI and Leaf Area Index, or (**C**) Tree Canopy and Leaf Area Index at 300 m buffers. NDVI = normalized difference vegetation index. LAI = Leaf area index. TC = tree canopy. Significance was set at *p* ≤ 0.05.

**Table 1 ijerph-19-16317-t001:** Demographic characteristics of the perceived greening participants stratified by Low/Medium/High NDVI values (n = 175) within 300-m-radius circular zone surrounding residence.

	Total	Low NDVI (0.00 to 0.36)	Medium NDVI (0.37 to 0.40)	High NDVI (>0.40 to 0.47)	*p* Value
Study Population	175	59	58	58	
Male	71 (40.6)	29 (49.2)	19 (32.8)	23 (39.7)	0.193
Race					0.510
White	142 (81.1)	50 (84.8)	44 (75.9)	48 (82.8)	
Black	24 (13.7)	5 (8.5)	11 (19.0)	8 (13.8)	
Other	9 (5.1)	4 (6.8)	3 (5.2)	2 (3.5)	
Hispanic	7 (4.0)	4 (6.9)	2 (3.5)	1 (1.7)	0.353
Income					0.003
less than $20,000	30 (17.1)	12 (20.3)	10 (17.2)	8 (13.8)	
$20,000–$64,999	89 (50.9)	32 (54.2)	37 (63.8)	20 (34.5)	
$65,000–$124,999	49 (28.0)	13 (22.0)	9 (15.5)	27 (46.6)	
Above $125,000	3 (1.7)	1 (1.7)	0 (0.0)	2 (3.5)	
Missing	4 (2.3)	1 (1.7)	2 (3.5)	1 (1.7)	
Education					0.097
≤High School Graduate	46 (26.3)	15 (25.4)	19 (32.8)	12 (20.7)	
2–4-year degree	98 (56.0)	37 (62.7)	31 (53.5)	30 (51.7)	
Master’s or Doctorate	30 (17.7)	7 (11.9)	7 (12.1)	16 (27.6)	
Missing	1 (0.6)	0 (0.0)	1 (1.7)	0 (0.0)	
Mother’s Education					0.250
≤High School Graduate	112 (64.0)	40 (67.8)	40 (69.0)	32 (55.2)	
2–4-year degree	44 (25.1)	11 (18.6)	12 (20.7)	21 (36.2)	
Master’s or Doctorate	10 (5.7)	4 (6.8)	3 (5.2)	3 (5.2)	
Missing	9 (5.1)	4 (6.8)	3 (5.2)	2 (3.5)	
Diabetes	45 (25.7)	19 (32.2)	15 (25.9)	11 (19.0)	0.261
Ever Smoked at least 100 Cigarettes	91 (52.0)	33 (55.9)	33 (56.9)	25 (43.1)	0.196
Age (Years)	50.84 (12.24)	51.04 (12.51)	51.69 (11.80)	49.78 (12.52)	0.698
Body Mass Index (kg/m^2^)	29.49 (6.01)	30.68 (5.89)	29.41 (5.78)	28.36 (6.25)	0.112
Systolic Blood Pressure (mmHg)	118.22 (16.89)	119.93 (15.10)	120.02 (19.83)	114.59 (14.93)	0.146
Diastolic Blood Pressure (mmHg)	79.33 (12.08)	80.60 (9.99)	79.95 (14.75)	77.36 (10.90)	0.320
Weight (Pounds)	187.31 (40.36)	197.47 (40.43)	185.64 (39.91)	178.65 (39.11)	0.038
Waist Circumference (Inches)	40.14 (6.09)	41.58 (6.14)	40.22 (5.65)	38.54 (6.18)	0.027
Hip Circumference (Inches)	43.54 (5.36)	44.21 (5.16)	43.76 (5.22)	42.61 (5.68)	0.261
Percentage Body Fat (%)	34.12 (10.64)	34.72 (9.86)	34.78 (10.85)	32.87 (11.25)	0.560

Frequencies and percentages were reported for categorical variables; means and standard deviations were reported for continuous variables. Differences in participant characteristics by tertiles of greenness metrics were tested using Chi-square test for categorical variables and one-way ANOVA for continuous variables. Significance was set at *p* ≤ 0.05.

**Table 2 ijerph-19-16317-t002:** Spearman Correlations between Individual Items with NDVI 300 m, LAI 300 m and Tree Canopy 300 m. n = 175.

Variable	NDVI 300 m	LAI 300 m	TC 300 m
**Item 1**	0.038	0.012	0.007
**Item 2**	0.140	0.220 *	0.174 *
**Item 3**	0.201 *	0.177 *	0.144
**Item 4**	0.129	0.071	0.072
**Item 5**	0.179 *	0.150 *	0.103
**Item 6**	0.258 *	0.230 *	0.210 *
**Item 7**	0.151 *	0.242 *	0.176 *
**Item 8**	0.164 *	0.258 *	0.194
**Item 9**	0.154 *	0.246 *	0.188 *
**Item 10**	0.166 *	0.251 *	0.196 *
**Item 11**	0.235 *	0.300 *	0.250 *
**Item 12**	0.201 *	0.194 *	0.134
**Item 13**	0.095	0.078	0.039
**Item 14**	0.093	0.151 *	0.104
**Item 15**	0.119	0.157 *	0.100
**Item 16**	0.167 *	0.247 *	0.176 *
**Item 17**	−0.086	−0.110	−0.070

NDVI = normalized difference vegetation index. LAI = leaf area index. TC = tree canopy. **Item 1**: Formal gardens nearby (e.g., botanical garden); **Item 2:** Children’s playground nearby; **Item 3:** Local park or nature reserve close to where I live; **Item 4:** Sports fields nearby (e.g., football/baseball field); **Item 5**: School grounds with grassed areas nearby; **Item 6:** Views of nature from my home; **Item 7:** Tree cover or canopy along walking routes; **Item 8:** Many large trees in my local area; **Item 9:** Many roadside plantings of trees and shrubs; **Item 10:** Many street trees in my local area; **Item 11:** Walking or bicycle paths or trails nearby; **Item 12:** Pleasant natural features (reserves, beach, lake); **Item 13:** Pockets of natural plants or native vegetation; **Item 14:** Lots of green median strips; **Item 15:** Lot of vegetation in nearby gardens; **Item 16:** Lots of greenery around my local area (trees, bushes, gardens); **Item 17:** Little or no lawn in nearby yards. * Represents significant correlation *p* ≤ 0.05.

**Table 3 ijerph-19-16317-t003:** Principal component analysis varimax rotated components.

	PC1 Natural Vegetation and Spaces	PC2 Trees	PC3 Playgrounds, Parks, and Sports Fields
**Eigenvalue**	7.26	1.76	1.32
**% Variance**	42.71	10.34	7.77
**Variables**			
** Formal gardens nearby (Item 1)**	0.797		
** Children’s playground (Item 2)**		0.267	0.777
** Local parks (Item 3)**		0.236	0.788
** Sports field (Item 4)**			0.779
** School ground/grasses (Item 5)**	0.337		0.557
** Views of nature (Item 6)**	0.515	0.355	0.298
** Tree cover or canopy (Item 7)**	0.247	0.749	0.220
** Many large trees (Item 8)**		0.768	0.269
** Roadside trees/scrubs (Item 9)**	0.492	0.651	
** Many street trees (Item 10)**	0.304	0.724	
** Walking bike paths/trails (Item 11)**	0.411	0.326	0.461
** Natural features (Item 12)**	0.806		
** Native vegetation (Item 13)**	0.712	0.227	0.337
** Green median strips (Item 14)**	0.658	0.398	
** Lots of vegetation/gardens (Item 15)**	0.769		
** Lots of greenery (Item 16)**	0.609	0.555	
** Little or no lawn in yards (Item 17)**		−0.498	

Item 1: Formal gardens nearby (e.g., botanical garden); Item 2: Children’s playground nearby; Item 3: Local park or nature reserve close to where I live; Item 4: Sports fields nearby (e.g., football/baseball field); Item 5: School grounds with grassed areas nearby; Item 6: Views of nature from my home; Item 7: Tree cover or canopy along walking routes; Item 8: Many large trees in my local area; Item 9: Many roadside plantings of trees and shrubs; Item 10: Many street trees in my local area; Item 11: Walking or bicycle paths or trails nearby; Item 12: Pleasant natural features (reserves, beach, lake); Item 13: Pockets of natural plants or native vegetation; Item 14: Lots of green median strips; Item 15: Lot of vegetation in nearby gardens; Item 16: Lots of greenery around my local area (trees, bushes, gardens); Item 17: Little or no lawn in nearby yards.

**Table 4 ijerph-19-16317-t004:** Association between principal component scores with NDVI, tree canopy, and leaf area in adjusted models. Beta per IQR of greenness exposure. NDVI IQR = 0.053. Tree canopy IQR = 7.827. Leaf area IQR = 0.278.

Outcome	Exposure	Adjusted β	95% CI	*p*-Value	Adjusted *p*-Value
PC 1	NDVI	0.161	−0.008, 0.330	0.062	0.111
PC 2	NDVI	0.088	−0.085, 0.262	0.316	0.355
PC 3	NDVI	0.265	0.092, 0.438	0.003	0.026
PC 1	Tree Canopy	0.194	−0.027, 0.415	0.084	0.122
PC 2	Tree Canopy	0.300	0.078, 0.523	0.009	0.026
PC 3	Tree Canopy	0.296	0.068, 0.524	0.011	0.026
PC 1	Leaf Area Index	0.085	−0.116, 0.286	0.404	0.404
PC 2	Leaf Area Index	0.173	−0.030, 0.376	0.095	0.122
PC 3	Leaf Area Index	0.280	0.075, 0.485	0.008	0.026

Values represent adjusted Betas and 95% confidence intervals per IQR of exposure. Principal Component scores were regressed against greenness metrics, adjusting for age, sex, race, income, education, and BMI. PC 1 = Natural vegetation and spaces. PC 2 = Trees. PC 3 = Playgrounds, parks, and sports fields. NDVI = normalized difference vegetation index. Adjusted *p*-value represents False Discovery rate adjusted *p*-value using the Benjamini-Hochberg procedure. The adjusted alpha was set at *p ≤* 0.028. Significance was set at Adjusted *p* ≤ 0.05.

## Data Availability

The data presented in this study are available on request from the corresponding author. The data are not publicly available due to privacy issues.

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
