# Peer review of "The Relationship between Perceptions and Objective Measures of Greenness"

_ijerph, 2022, doi:10.3390/ijerph192316317_

Round 1

Reviewer 1 Report

The authors investigated the correlation between objective measures of greenness and human perceptions of greenness. This study is interesting and seems to be suitable for the potential readers of this journal. However, before publishing this research article, some major and minor points should be addressed. 

1. The authors should include meta-review process and reliable scientific rationale for the measures since the literature survey on measures for human perceptions of greenness is too short and lacks of reliable discussion pertinent to why the objective measures described in the material section are valid for investigating the human perception correlates of greenness. In addition, this kind of literature survey and valid hypotheses on the proposed measures should be described before stating the purpose of the study. 

2. I suggest 3.1  Study Characteristics and Table 1 (demographic characteristics of the perceived greening participants) can be moved in 2.1 participants in the Methods. Statistical abbreviations (e.g., p and r) should be always in italics.

3. False discovery rates caused by multiple comparisons of PCA results should be addressed and inflated alpha must be controlled and corrected. I suggest to report the corrected alpha level and handle the FDR (alpha level should be divided by number of comparisons). Or the authors can consider applying BH methods to this results. Refer to the following reference (Effects of mental workload on involuntary attention: A somatosensory ERP study, Neuropsychologia 106, 7-20, 2017).

4. All vertical lines in Tables should be deleted for readability.

5. Result section is too short and I suggest the Supplemental Table 2. is included in the results. Additional analytics of the presented data should be made to support the conclusion. 

6. Practical significance of the spearman correlations between Individual Items with NDVI 300m, LA 300m and Tree Canopy 300m should be discussed. Only statistical significant indices were just reported. Even though r of 0.151 is statistically significant, whether the coefficient is practically meaningful should be discussed and any possible interpretations underlying the results need to be presented for potential readers.  

7. The authors stated that ANOVA for continuous variables was conducted. What kinds of ANOVAs (e.g., one-way ANOVA, two-way repeated measures of ANOVA, and etc.) was conducted should be clearly described. In addition, I found no results related to the ANOVAs (i.e., F, degree of freedom, and eta squared).  

Reviewer 2 Report

Dear Authors,

This paper presents the relationship between perceptions and objective measures of greenness.

Research include data collection methods, results, discussion and conclusions. I appreciate the interdisciplinary nature of research and the cooperation of several universities - both urban planners, landscape architects and doctors. A very important problem was examined. I think this is a good article. Figures, charts and tables are properly prepared. The conclusions were properly presented.

My feeling is that it should be published with the following suggestion taken into account. 

1.      I suggest adding more literature (there should be at least 50 items in the reference list) e.g.:

And they should be adapt your References to MDPI.  https://www.mdpi.com/journal/ijerph/instructions

1.       Kor, P.P.K.; Liu, J.Y.W.; Chien, W.T. Effects of a modified mindfulness-based cognitive therapy for family caregivers of people with dementia: A randomized clinical trial. Gerontologist 202161, 977–990. [Google Scholar] [CrossRef]

      Yours sincerely,

Reviewer.

Round 2

Reviewer 1 Report

Thank you for your revisions to the manuscript. Most of the issues have been addressed, except for the following part. The authors applied the BH method, but applied the method in the p values reported. To correct inflated type I error caused by multiple comparisons, the authors should apply the adjusted alpha level to judge their significance. For example, if the authors determined the adjusted alpha as 0.006 by the BH method, the null hypotheses with p values above the adjusted alpha level should not be rejected. The authors may use the following expression : trending toward significance (if some p-values are between 0.006 and 0.05.  

Reviewer 2 Report

Dear Authors. Thank you for your revisions to the manuscript. I appreciate all works done by Authors. Most of the issues have been addressed.  Bibliography - it could be developed further but in present form is acceptable.  
